# Characterization of Protein Hydrolysates from Fish Discards and By-Products from the North-West Spain Fishing Fleet as Potential Sources of Bioactive Peptides

**DOI:** 10.3390/md19060338

**Published:** 2021-06-13

**Authors:** Andreia Henriques, José A. Vázquez, Jesus Valcarcel, Rogério Mendes, Narcisa M. Bandarra, Carla Pires

**Affiliations:** 1Division of Aquaculture and Upgrading and Biospropecting (DivAV), Portuguese Institute for the Sea and Atmosphere (IPMA, I.P.), Av. Dr. Alfredo Magalhães Ramalho 6, 1495-165 Lisboa, Portugal; andreia.henriques@ipma.pt (A.H.); rogerio@ipma.pt (R.M.); narcisa@ipma.pt (N.M.B.); 2Group of Recycling and Valorization of Waste Materials (REVAL), Marine Research Institute (IIM-CSIC), R/Eduardo Cabello 6, 36208 Vigo, Spain; jvazquez@iim.csic.es (J.A.V.); jvalcarcel@iim.csic.es (J.V.); 3Interdisciplinary Center of Marine and Environmental Research (CIIMAR), Terminal de Cruzeiros de Leixões, Av. General Norton de Matos s/n, 4450-208 Matosinhos, Portugal

**Keywords:** bioactive peptides, radical scavenging activity, Fe^2+^ chelating activity, α-amylase and α-glucosidase inhibitory activities, ACE inhibitory activity

## Abstract

Fish discards and by-products can be transformed into high value-added products such as fish protein hydrolysates (FPH) containing bioactive peptides. Protein hydrolysates were prepared from different parts (whole fish, skin and head) of several discarded species of the North-West Spain fishing fleet using Alcalase. All hydrolysates had moisture and ash contents lower than 10% and 15%, respectively. The fat content of FPH varied between 1.5% and 9.4% and had high protein content (69.8–76.6%). The amino acids profiles of FPH are quite similar and the most abundant amino acids were glutamic and aspartic acids. All FPH exhibited antioxidant activity and those obtained from Atlantic horse mackerel heads presented the highest 2,2-diphenyl-1-picrylhydrazyl (DPPH) radical scavenging activity, reducing power and Cu^2+^ chelating activity. On the other hand, hydrolysates from gurnard heads showed the highest ABTS radical scavenging activity and Fe^2+^ chelating activity. In what concerns the α-amylase inhibitory activity, the IC_50_ values recorded for FPH ranged between 5.70 and 84.37 mg/mL for blue whiting heads and whole Atlantic horse mackerel, respectively. α-Glucosidase inhibitory activity of FPH was relatively low but all FPH had high Angiotensin Converting Enzyme (ACE) inhibitory activity. Considering the biological activities, these FPH are potential natural additives for functional foods or nutraceuticals.

## 1. Introduction

According to the Food and Agriculture Organization [1] the total global fish production reached about 179 million tonnes in 2018, 88% of it (156 million tonnes) being used for direct human consumption. The remaining 22 million tonnes were destined for non-food purposes, of which 12 million tonnes were converted into fishmeal and fish oil. In relation to underutilized fish resources, there are also significant amounts of discards on board fishing vessels, which represent 9–15% of total global catches [1]. Moreover, the 2013 Reform of the Common Fisheries Policy and the introduction of the Landing Obligation, which requires all catches of regulated commercial species to be landed and counted against quota, is leading to an increase in fish landed that cannot be used in direct human consumption. In addition to this landed fish, there are also large amounts of by-products from the fish processing factories, which may represent up to 70% of processed fish and can also cause significant environmental problems. These by-products are usually composed of heads (9–12% of total fish weight), viscera (12–18%), skin (1–3%), bones (9–15%) and scales (about 5%) [2]. Reports show that fishery products have been traditionally used in some cultures in the production of animal-derived medicinal products for treatment of exhaustion and several diseases (e.g., colds, pneumonia, burns) [3]. Nowadays, it has been envisaged as a source of bioactive compounds with medical applications, among others.

Proteins of discarded species and by-products can be enzymatically hydrolyzed to produce fish protein hydrolysates (FPH). These hydrolysates contain unique bioactive peptides having a wide range of biological activities, which could be used as active components in modern drugs. In fact, FPH exhibit different activities such as antioxidant, anti-hypertensive, antimicrobial, antitumoral, anti-inflammatory, antidiabetic and other activities such as calcium binding and anticoagulant [4,5]. The amino acid sequences of many bioactive fish peptides have been identified; however, there is still a lack of pre-clinical and pharmacokinetic studies on the use of these peptides [6].

The antioxidant activity of FPH has been widely studied by several authors because they may be an alternative to synthetic antioxidants. FPH exhibiting antioxidant activity have been prepared from different parts of fish including viscera [7], backbone [8], skin [7] and muscle [9]. The prolonged use of synthetic inhibitors of Angiotensin Converting Enzymes (ACE) such as captopril, enalapril, alacepril and lisinopril can causes adverse effects [10]. Thus, the interest in finding natural inhibitors has increased. Peptides from oyster proteins [11], shark meat [12], salmon by-products [13], leatherjacket mince [14], walleye pollock, yellowfin sole, cod bones and cod heads [15] were shown to significantly inhibit ACE.

Furthermore, several studies on the anti-diabetic activity of peptides from milk, soy and peas or lupine have also been reported [16]. However, a limited number of studies have been published on the anti-diabetic activity of fish protein hydrolysates/peptides through the inhibition of α-amylase and α-glucosidase. Likewise, there is a shortage of information regarding the potential of new sources of bioactive peptides from by-products and discards, deriving from the landing obligation in some fishing areas. Therefore, the aim of the current work was to characterize (proximate, amino acid and organic matter compositions, digestibility, molecular weight distribution) and evaluate the biological activities (antioxidant, anti-hypertensive and anti-diabetic) of hydrolysates obtained from different parts (whole fish (Wh), skins and bones (Sb) and heads (He)) of the most recurrently discarded fish species in trawler fisheries in North-West Spain, namely blue whiting (BW, *Micromesistius poutassou*), Atlantic horse mackerel (AHM, *Trachurus trachurus*), gurnard (Gu, *Trigla* spp.), pouting (Po, *Trisopterus luscus*), red scorpionfish (RS, *Scorpaena scrofa*) and four spot megrim (M, *Lepidorhombus boscii*).

## 2. Results and Discussion

### 2.1. Chemical Composition of FPH

The main chemical characteristics of FPH are summarized in Table 1. Moisture and ash contents were lower than 10% and 15%, respectively, in all FPH whereas organic matter ranged from 79% to 83%. Several authors have reported moisture content for various FPH below 10% [17]. The relatively high ash content of FPH (11.9–15.4%) is due to the addition of NaOH for pH adjustment during the hydrolysis process. The reported ash content for FPH prepared with different fish species ranged between 0.45–27% [17]. Ash content of protein hydrolysates primarily depends on the hydrolysis process.

The fat content of FPH varied between 1.5% (RS_Wh) and 9.4% (BW_He) and these differences are related to the type of raw material and species used to prepare the FPH. Most of the reported studies have reported a variety of FPH with a fat content lower than 5%, whereas only a few studies mentioned FPH with fat content above 5% [17]. The relatively low-fat content obtained in the hydrolysates is due to the removal of lipids released in the hydrolytic process. FPH from Wh and Sb presented the highest protein concentration (76.6% in Po_Sb). Conversely, FPH prepared from heads showed the lowest protein levels (70.1–71.9%). The differences observed in protein content of the different FPH were mainly related to the variation of the fat content. However, the protein content of these hydrolysates was of the same order of magnitude of those prepared from different species [17].

The gel permeation profile of all FPH indicated hydrolysis of proteins into peptides with small molecular weights (Mn < 1800 Da) and free amino acids. The highest peptides size, both in number average molecular weight (Mn) and average molecular weight (Mw), were determined in hydrolysates prepared from heads, whereas FPH prepared from skins presented peptides with the lowest Mn and Mw (Table 1). Variation in the hydrolysates molecular weight profiles can be explained by differences in protein content of raw material and type of protein substrate used in hydrolysis, as mentioned by Barkia et al. [18].

The in vitro digestibility of FPH was in all cases excellent with values higher than 89.7%, reaching 97.2% in BW_Wh (Table 1). The FPH prepared from heads presented the lowest protein digestibility values (89.7–92.3%).

The amino acid profiles of the different hydrolysates are quite similar (Table 2 and Table 3) and contained all the essential amino acids. Glutamic (12–15%) and aspartic acid (8–10%) were the most important amino acids. In fact, these amino acids have been shown to present the highest concentration in the majority of the fish protein hydrolysates [17].

In general, the percentage of the different amino acids was lower in hydrolysates prepared with Sb and the order was Sb < He < Wh, apart from glycine, proline and alanine, where this order is reversed. The differences in the levels of amino acids between the different hydrolysates was related to the type of raw material used in the hydrolysis. The higher levels of glycine and proline in Sb hydrolysates indicate the presence of a large amount of collagen in this raw material.

Regarding the ratio of total essential amino acids/total amino acids (TEAA/TAA), the results were in the interval of 29.76% (Gu_He) and 42.82% (RS_Wh) with the highest ratios in hydrolysates prepared with whole fish, followed by FPH prepared with Sb and He. The presence of a higher content of muscle substrate when the whole fish was used in the preparation of FPH may justify the difference.

### 2.2. Antioxidant Activity of FPH

#### 2.2.1. DPPH Scavenging Activity

DPPH has been widely used to estimate the free radical scavenging activities of antioxidants. The DPPH radical scavenging capacity of FPH prepared with different fish discards is shown in Table 4. All hydrolysates exhibited a DPPH scavenging activity with a concentration dependence, but 50% of inhibition was not achieved in the concentration range tested. The scavenging activity was significantly higher in the hydrolysates prepared from heads (He) of AHM and RS. In contrast, FPH prepared from skins and bones (Sb) showed the lowest scavenging activity, with the exception of that prepared from AHM. This is in accordance with the peptide profile of AMH hydrolysates and the presence of lower molecular weight peptides in these hydrolysates (Table 1), since it has been reported that lower molecular weight peptides exhibited higher DPPH scavenging activity [19,20,21].

The relatively low inhibition percentage values obtained in the current work were similar to those reported for hydrolysates prepared from hake muscle [9], which presented ca. 18% inhibition for a 5 mg/mL hydrolysate concentration. However, protein hydrolysates obtained from sardinella heads and visceras [18] and round scad muscle [22] showed high DPPH radical scavenging activity. On the other hand, reported data show an inhibition of 80% in yellow trevally protein hydrolysates (40 mg/mL) produced using Alcalase and Flavourzyme [23]. These differences may result from the raw material [24], hydrolysis conditions [25] or the enzymes used in their preparation [26].

#### 2.2.2. ABTS^•+^ Radical Scavenging Activity

The ABTS radical scavenging assay measures the potential of an antioxidant to inhibit the ABTS radical cation [27]. The EC_50_ values of FPH varied between 1.12 and 4.93 mg/mL (Table 4). The EC_50_ (mg/mL) values of G_Wh (1.47 ± 0.020 mg/mL) and G_He (1.12 ± 0.026 mg/mL) were significantly lower than those of the other FPH. On the other hand, Atlantic horse mackerel hydrolysates (AHM_He, AHM_Sb and AHM_Wh) showed the highest EC_50_ values.

The radical scavenging activity of fish protein hydrolysates obtained in the current work was similar to those prepared from hake by-products by Teixeira et al. [28], who found EC_50_ values of 2.3 mg P/mL. Reported EC_50_ values of hydrolysates prepared from panga myofibrillar proteins, with Alcalase and Flavourzyme, were 0.893 ± 0.31 mg/mL and 1.490 ± 0.23 mg/mL, respectively [29]. Protein hydrolysates of common carp roe [30] and blue-spotted stingray [31] also had a greater ABTS inhibitory activity (EC_50_ = 0.301 mg/mL and EC_50_ = 0.79 mg/mL, respectively) than those prepared in this study. Bkhairia et al. [32] evaluated the ABTS scavenging activity of mullet muscle protein hydrolysates prepared with different proteases and EC_50_ values obtained varied between 0.47 mg/mL and 0.81 mg/mL. On the other hand, hydrolysates prepared from tilapia muscle using Alcalase and Flavourzyme exhibited 91.27% and 88.13% of inhibition, respectively, for a 66.67 μg/mL hydrolysate concentration [33].

In summary, the FPH showed higher radical scavenging capacity of ABTS than DPPH, which has been observed by other authors and suggest that these hydrolysates are richer in hydrophilic peptides [34].

#### 2.2.3. Reducing Power

The reducing power assay (RP) has been used to evaluate the ability of an antioxidant to donate electrons [35].

A linear increase in reducing power with the concentration of hydrolysates was observed for all hydrolysates. Other authors also obtained a similar trend between the concentration of fish protein hydrolysates and RP [8,9,36,37,38]. To compare the RP capacity of the FPH prepared and the results reported by other authors, the A_0.5_ value was used. According to Zhou et al. [39], this value is the concentration of a sample required to produce an absorbance of 0.5. The A_0.5_ value of FPH ranged from 3.19 to 6.35 mg/mL (Table 4). The lowest A_0.5_ value was recorded for AHM_He (3.19 ± 0.057 mg/mL) and the highest was obtained with the hydrolysate prepared from M_He (6.35 ± 0.037 mg/mL).

The results of this study were lower than those reported by several authors. For example, Zhou et al. [39] reported A_0.5_ values of 11.0 mg/mL and 11.5 mg/mL for hydrolysates prepared from abalone foot muscle and scallop adductor muscle, respectively. An A_0.5_ value of the same order of magnitude (ca 15.0 mg/mL) was obtained by Pires et al. [9] for Cape hake protein hydrolysates, while García-Moreno et al. [37] referred A_0.5_ values in the range 10–31.25 mg/mL for FPH prepared from different species. In turn, the results obtained by Ktari et al. [36] pointed out to an A_0.5_ value of ca. 2.5 mg/mL for zebra blenny muscle hydrolysates, whereas about 5.5 mg/mL was reported for hydrolysates from pink perch muscle [8]. The RP of protein hydrolysates of roe carp (A_0.5_ ≈ 2 mg/mL) obtained by Chalamaiah et al. [30] was also lower than the determined in the present study. Chen et al. [40] also reported higher RP for protein hydrolysates prepared from tilapia sarcoplasmic proteins with papain, where a 3 mg/mL hydrolysate solution showed an absorbance of 0.629 ± 0.022.

### 2.3. Metal Chelating Activity of FPH

#### 2.3.1. Cu^2+^ Chelating Activity

Transition metal ions such as Cu^2+^ and Fe^2+^ can catalyze the generation of reactive oxygen species, which may lead to lipid peroxidation and DNA damage [41]. The EC_50_ values of Cu^2+^ chelating capacity of the FPH prepared in the current work are shown in Table 4. The EC_50_ values of FPH varied between 2.49 and 5.66 mg/mL and the Atlantic horse mackerel hydrolysates (AHM_He and AHM_Wh) and megrim hydrolysates (M_He and M_Wh) had the lowest values. Conversely, blue whiting hydrolysates presented significantly higher EC_50_ values.

The different FPH had EC_50_ values similar to those referred by You et al. [42] for protein hydrolysates prepared from loach muscle with papain (EC_50_ = 2.89 ± 0.01 mg/mL) and Chai et al. [31] for hydrolysates obtained from Blue-spotted Stingray muscle with Alcalase (EC_50_ = 2.14 ± 0.014 mg/mL). On the other hand, the Cu^2+^ chelating capacity of the FPH prepared in the current study were lower than that of hydrolysates prepared from Cape hake by-products by Teixeira et al. [27] (EC_50_ = 1.4 ± 0.1 mg/mL).

The Cu^2+^ chelating activity of the different hydrolysates could be related to the presence of high content of carboxylic acids (Asp and Glu) which are mainly responsible for the chelating activity of protein hydrolysates as mentioned by Zhu et al. [43].

#### 2.3.2. Fe^2+^ Chelating Activity

The chelating capacity of peptides depends not only on their size, but also on the amino acids’ composition and respective sequence in the peptides. For example, peptides with histidine show a strong chelating ability of metal ions due to the presence of the imidazole ring [17]. Such as in the Cu^2+^ chelation, the amino acids involved in the Fe^2+^ chelation are also Asp, Glu and His, as reported by Kong and Xiong [44].

The Fe^2+^ chelating ability of FPH is shown in Table 4. All hydrolysates showed higher Fe^2+^ chelating activity than Cu^2+^ chelating activity, as evidenced by the lower EC_50_ values achieved (0.26–0.53 mg/mL). In general, the EC_50_ values of all hydrolysates were similar but FPH prepared from skins and bones presented lower EC_50_. However, AHM_He (heads) showed the highest EC_50_ value.

These values were of the same order of magnitude of those referred by Kumar et al. [7] for hydrolysates prepared from skins of horse mackerel and croaker. Likewise, reported data from viscera of black pomfret [45], from several Mediterranean fish species [37] and from Cape-hake by-products [28], showed EC_50_ values of the same magnitude. However, higher EC_50_ values were obtained in hydrolysates prepared from yellow stripe trevally [23] and in tuna liver hydrolysates [46]. On the other hand, hydrolysates prepared from blue-spotted stingray presented lower EC_50_ [31] than those prepared in this study.

Intarasirisawat et al. [47] showed that the chelating capacity of Fe^2+^ of tuna eggs protein hydrolysates increased with the degree of hydrolysis. According to these authors, the stereochemical impediment of peptides with a larger average size could decrease the ability to migrate and chelate the target metal ion. However, in the current study, the hydrolysates prepared from skins and bones had the lowest average peptides size and the lowest Fe^2+^ chelating capacity.

FPH hydrolysates prepared in this work exhibited higher chelating activity for Fe^2+^ than for Cu^2+^.

### 2.4. Anti-Diabetic Activity

#### α-Amylase and α-Glucosidase Inhibitory Activities

The inhibition of α-amylase and α-glucosidase activities has been considered a significant approach for controlling obesity. The hydrolyzed dietary carbohydrates are the main source of increased level of glucose in blood. Thus, the inhibition of amylases and glucosidases would avoid complex polysaccharides from becoming hydrolyzed and then absorbed in bloodstream. The anti-obesity activity of FPH prepared with different species and by-products was evaluated by the inhibitory activity of these two enzymes.

The α-amylase inhibitory activity of the FPH was measured as a function of hydrolysate concentration in order to achieve the half maximal inhibitory concentration (IC_50_). As shown in Table 5, there were big differences in the inhibitory capacity among FPH, as evidenced by the IC_50_ values recorded which ranged between 5.70 and 84.37 mg/mL. Nevertheless, an increase in this activity with increasing hydrolysate concentration was observed. Among FPH, BW_He and BW_Sb showed the highest α-amylase inhibitory activity. In contrast, Po_He and AHM_Wh exhibited a significantly lower α-amylase inhibitory activity and in the case of Po_Wh and RS_He, 50% of inhibition was not attained in the range of concentrations tested. These big differences in the inhibitory activity among FPH may be attributed, as in the case of other biological activities, to the smaller size and amino acid composition of hydrolysate peptides. The presence of Gly or Phe at the N-terminal and Phe or Leu at the C-terminal in the peptide chains affected the α-amylase inhibitory activity as reported by Ngoh and Gan [48].

A limited number of works on the α-amylase inhibitory activity of fish protein hydrolysates have been published. Siala et al. [49] followed a different methodology to measure the inhibition of α-amylase by grey triggerfish muscle hydrolysates and very low IC_50_ values, in the range of 90–93 μg/mL, were reported. In addition, Salem et al. [50], in a study on the α-amylase inhibition by hydrolysates of octopus prepared with different enzymes, obtained values of IC_50_ between 61.34 μg/mL for the hydrolysate prepared with Esperase and 66.22 μg/mL for the hydrolysate obtained with Bacillus subtilis A26. Low α-amylase inhibition was also observed for silver warehou, barracouta and Australian salmon hydrolysates [51].

In what concerns α-glucosidase inhibitory activity a concentration-dependent activity was observed for all protein hydrolysates at concentrations between 25 and 200 mg/mL. However, 50% inhibition was not achieved for BW_He and BW_Sb hydrolysates. The IC_50_ values ranged between 21.8 (RS_Sb) and 300 mg/mL (RS_Wh).

A study with sardine muscle hydrolysates showed inhibitory activity against α-glucosidase with an IC_50_ value of 48.7 mg/mL when PNP-G was used as substrate [52]. On the other hand, Medenicks and Vasiljevic [51] reported that no α-glucosidase inhibition was observed for silver warehou, barracouta and Australian salmon hydrolysates. Like α-amylase inhibitory activity, α-glucosidase inhibitory of fish peptides has also not been broadly studied. Thus, the FPH inhibitory activity was compared with that obtained on protein hydrolysates from other types of proteinous raw material. For example, Wang et al. [53] reported an IC_50_ value of 4.94 ± 0.07 mg/mL for α-glucosidase inhibitory activity of soy protein hydrolysates. The α-glucosidase inhibitory activity of pea protein hydrolysates was 38.39 ± 1.58% for a 20 mg/mL hydrolysate concentration [54]. On the other hand, using the same hydrolysate concentration, Karimi et al. [55] reported an inhibitory activity of corn germ protein hydrolysates prepared with enzymes, on the range of 12.8 and 37.1%. Arise et al [56] reported strong α-glucosidase inhibitory activity (65.81 ± 1.95%) for a *Luffa cylindrica* seeds hydrolysates (1 mg/mL).

### 2.5. ACE Inhibitory Activity

The ability of hydrolysates to inhibit the ACE is shown in Table 5. All FPH with 5 mg/mL concentration exhibited high ACE inhibitory activity (61.20–85.95%). In general, FPH prepared from heads had lower ACE inhibitory activity (60.77–79.35%). On the other hand, the highest inhibition percentage was achieved with M_Wh. It is noteworthy that hydrolysates prepared from heads had peptides with larger molecular sizes. This agrees with the results reported by Bougatef et al. [57] who observed an increase in ACE inhibition with increasing degree of hydrolysis of sardinella (*S**ardinella aurita*) protein by-products.

The ACE inhibitory activity percentage obtained in the current study is similar to the reported by other authors [58,59,60,61]. However, other authors referred IC_50_ values of 0.34–0.41 mg/mL for salmon gelatin hydrolysates prepared with Alcalase [62] and 0.34 mg/mL for squid gelatin hydrolysates [63].

It is well known that the IC_50_ value of fish protein hydrolysates depends on several parameters such as enzyme and substrate used in the hydrolysis and hydrolysis conditions. These factors will lead to the formation of peptides with different molecular weights and amino acid sequences that ultimately will be responsible for the differences in the ACE inhibitory activities achieved.

### 2.6. Principal Components Analysis

A multivariate analysis of the relationship between data obtained from amino acids profile and antioxidant activities of FPH was conducted in order to detect groups of samples (Figure 1). The results show that the first principal component (PC1, 36.46% of the total explained variance) is strongly correlated with Leu (loading 0.93), Ile (0.93), Gly (−0.96), Ala (−0.87), Val (0.86), Pro (−0.85) and Lys (0.83). The second PC (PC2, 13.57% of the total explained variance) is correlated with ACE inhibitory activity, DPPH radical scavenging activities, Fe^2+^ chelating activity and Mw (loadings were 0.75, −0.59, −0.66, 0.57 respectively), while ABTS is correlated with the third PC. Cu^2+^ chelating activity is correlated with the fourth PC, but it is not plotted as it did not add information to the study.

The plot (PC1 vs. PC2) shows a clear separation of FPH prepared from the different raw material based on PC1, specially on Gly, Ala and Pro content. FPH prepared from skins and bones were richer in Pro, Ala and Gly than those prepared from heads and whole fish. On the other hand, FPH prepared from heads had higher levels of Met, Tir and Glu while FPH obtained from whole fish had higher levels of Hist, Thr, Cys, Phe, Leu, Lys, Val and IIe. FPH prepared from heads and whole fish are separated in two groups based on PC2. FPH prepared from heads had lower ACE inhibitory activity, but higher Mw and Mn, DPPH radical scavenging activity and Fe^2+^ chelating activity. The findings suggested by PCA were supported by ANOVA.

The proximity of the projection of variables in the plot of principal components (PC1 and PC2) suggests a correlation between DPPH radical scavenging activity and Fe^2+^ chelating activity (r = 0.73), DPPH and ACE inhibitory activity (r = −0.60) and between DPPH and Pro (r = −0.51). ACE inhibitory activity is correlated with Fe^2+^ chelating activity (r = −0.64), with α-amylase inhibitory activity (r = 0.55) and with RP (r = 0.50).

## 3. Material and Methods

### 3.1. Fish Materials

Fish protein hydrolysates (FPH) were obtained using, as substrates, some of the most discarded fish species by the North-West Spain fishing fleet, namely gurnard (Gu, *Trigla* spp.), Atlantic horse mackerel (AHM, *Trachurus trachurus*), blue whiting (BW, *Micromesistius poutassou*), red scorpionfish (RS, *Scorpaena scrofa*), pouting (Po, *Trisoreptus luscus*) and four-spot megrim (M, *Lepidorhombus boscii*). These species were captured in the North Atlantic Ocean (ICES areas VIIIc and IXa) in 2020, separated on board from commercial species and directly preserved in ice. After landing, a part of the fish was separated for processing as whole fish, and the rest of discards were manually headed and gutted. Headed and gutted fish were then processed in a meat-bone-skin separator (Josmar JM-301, Pontevedra, Spain) for fish mince production. Whole fish, heads and skins and bones from each mentioned fish species were subsequently homogenized by grinding and then stored at −18 °C until used as substrates for FPH.

### 3.2. Production of Fish Protein Hydrolysates

Hydrolysis was performed in a controlled pH-Stat system with a 5 L glass reactor mixing 1 kg of grinded discards with 2 L of distilled water (1:2). As an alkaline reagent for pH-control, 5 M NaOH was employed. The proteolysis step was carried out, in all cases, at 60 °C, pH 8.65, adding 1% (*v*/*w*) of Alcalase 2.4 L (2.4 Anson Unit/g, Novozymes, Nordisk, Bagsværd, Denmark), under continuous stirring at 200 rpm for 4 h [64]. The commercial endoprotease, Alcalase, was applied for the production of fish hydrolysates due to its well-known high proteolytic activity and efficiency to digest a broad type of marine wastes. After hydrolysis, the slurry was filtered (100 mm) to remove bones and the filtrate was centrifuged (15,000 *g* for 20 min) to separate oil and hydrolysates. Immediately, these FPH were heated (90 °C/15 min) for enzyme deactivation and dried by lyophilization for 72 h. To maintain their stability, dry FPH were subsequently vacuum-packed and stored at −18 °C until use.

### 3.3. Chemical Characterization of FPH

The yield of FPH production (Y in%, *v*/*w*) was calculated as the ratio between the volume of final FPH obtained after bones and oil separation and the sum in weight of the solid substrate, water and alkalis added for proteolysis. Moisture (Mo), ash and organic matter (OM) and total protein calculated as total nitrogen × 6.25 of FPH were determined following the AOAC methods [65]. Total fat content (TF) was determined according to Bligh and Dyer [66]. The in vitro FPH digestibility was done following the pepsin method (AOAC Official Method 971.09) with the modifications reported by Miller et al. [67].

### 3.4. Amino Acid Analysis

The amino acid profile was quantified by ninhydrin reaction, using an amino acid analyzer (Biochrom 30 series, Biochrom Ltd., Cambridge, UK), according to the method of Moore et al. [68].

### 3.5. Molecular Weight Analysis

The molecular weight distributions of FPH were obtained by Gel Permeation Chromatography (GPC,) [69]. The system (Agilent 1260 HPLC, Santa Clara, CA, USA) consisted of a quaternary pump, injector, column oven, refractive index, diode array and dual-angle light scattering detectors. The samples were eluted with 0.15 M ammonium acetate/0.2 M acetic acid (pH 4.5) at 1 mL/min after a 100 µL injection. Separation was achieved with a set of four Proteema columns (PSS GmbH, Mainz, Germany): precolumn (5 µm, 8 × 50 mm), 30 Å (5 µm, 8 × 300 mm), 100 Å (5 µm, 8 × 300 mm) and 1000 Å (5 µm, 8 × 300 mm) kept at 30 °C. Detectors were calibrated with a polyethylene oxide standard with an average weight molecular weight of 106 kDa (polydispersity index 1.05) from PSS (Mainz, Germany). Number average molecular weight (Mn) and average molecular weight estimations were conducted with refractive index increments (dn/dc) of 0.185.

### 3.6. Biological Activities

#### 3.6.1. DPPH Radical Scavenging Activity

The determination of DPPH radical scavenging activity was carried out according to the method of Shimada et al. [70] with some adjustments as described by Picot et al. [71]. One milliliter of the different FPH hydrolysates (1–5 mg/mL) was added and mixed with 1.0 mL of 0.1 mM DPPH solution in 95% ethanol in an *Eppendorf* tube. The solution was stored for 1 h at room temperature in the dark; thereafter, samples were centrifuged at 10,000 *g* for 10 min. The absorbance of the solution was measured at 517 nm using an Evolution 201 UV-Visible Spectrophotometer (Thermo Scientific, Waltham, MA, USA). The control was prepared using distilled water instead of the sample solution. The radical scavenging activity of FPH was calculated by the percentage inhibition of DPPH as follows:$${\mathrm{DPPH}}^{.}\;\mathrm{scavenging}\;\mathrm{activity}\;(\%)=\frac{{\mathrm{Abs}}_{\mathrm{control}}{-\mathrm{Abs}}_{\mathrm{sample}}}{{\mathrm{Abs}}_{\mathrm{control}}}\times 100$$
where Abs_sample_ and Abs_control_ correspond to the absorbance of sample and control, respectively. All analyses were made at least in triplicate and the results are presented as mean values.

#### 3.6.2. ABTS Radical Scavenging Activity

The ABTS radical scavenging activity of FPH was performed according to Re et al. [72]. ABTS radical cation ABTS^•+^ was prepared with a final concentration of 7 mM ABTS in 2.45 mM potassium persulfate. This mixture was kept in the dark at room temperature for 16 h before use. ABTS^•+^ solution was diluted with 5 mM sodium phosphate buffer (pH 7.4) to obtain an absorbance value of 0.70 ± 0.02 at 734 nm. A 20 µL aliquot of hydrolysates solution at different concentrations (0.5–10 mg/mL) was mixed with 2 mL of ABTS^•+^ solution and then incubated in the dark at 30 °C for 6 min. The absorbance values of the mixture were read at 734 nm using an Evolution 201 UV-Visible Spectrophotometer (Thermo Scientific). The control was prepared in the same manner using distilled water instead of the sample solution. All determinations were made at least in triplicate and the EC_50_ value was calculated for each hydrolysate. The ABTS scavenging activity was calculated according to the following equation:$${\mathrm{ABTS}}^{•+}\mathrm{scavenging}\;\mathrm{activity}\;(\%)=\frac{{\mathrm{Abs}}_{\mathrm{control}}{-\mathrm{Abs}}_{\mathrm{sample}}}{{\mathrm{Abs}}_{\mathrm{control}}}\times 100$$
where Abs_control_ represents the absorbance of the control and Abs_sample_ represents the absorbance of sample.

#### 3.6.3. Reducing Power

The reducing power was determined following Oyaizu’s method [73]. Two milliliters of hydrolysate solutions with different concentrations (1–10 mg/mL) were mixed with 2.0 mL phosphate buffer (0.2 M, pH 6.6) and 2.0 mL potassium ferricyanide 1%. After being incubated for 20 min at 50 °C, 2.0 mL of TCA 10% were added and centrifuged at 1500 *g* for 10 min. Finally, 2.0 mL of the supernatant solution were mixed with 2.0 mL distilled water and 0.4 mL of 0.1% ferric chloride (FeCl3) and the absorbance measured at 700 nm (Evolution 201 UV-Visible Spectrophotometer, Thermo Scientific), after 10 min incubation in the dark. The control was prepared using distilled water instead of sample solution. All analyses were carried out at least in triplicate. The concentration for the absorbance value of 0.5 value (A_0.5_) was determined for each hydrolysate.

#### 3.6.4. Cu^2^^+^ Chelating Activity

Copper chelating activity was evaluated by copper chelate titration using pyrocatechol violet (PV) as the metal chelating indicator [74], as described by Torres-Fuentes et al. [75] with slight modifications.

One milliliter of 0.1 mg/mL CuSO_4_ in 50 mM sodium acetate buffer pH 6.0 was mixed with 1 mL of sample solution prepared at different concentrations (0.1–2 mg/mL). Then, 250 µL of PV 0.3 mM in 50 mM sodium acetate buffer pH 6.0 were added and the PV+Cu^2+^ complex was formed. The absorbance was read at 632 nm (Evolution 201 UV-Visible Spectrophotometer, Thermo Scientific). The control was prepared in the same way by using distilled water instead of the sample solution. All determinations were carried out at least in quadruplicate and the EC_50_ was estimated for each hydrolysate. Chelating activity was calculated using the following formula:$$\mathrm{Copper}\;\mathrm{chelating}\;\mathrm{activity}\;(\%)=\frac{{\mathrm{Abs}}_{\mathrm{control}}{-\mathrm{Abs}}_{\mathrm{sample}}}{{\mathrm{Abs}}_{\mathrm{control}}}\times 100$$
where Abs_sample_ and Abs_control_ correspond to the absorbance of sample and control, respectively.

#### 3.6.5. Fe^2+^ Chelation Activity

The iron chelating activity of the FPH was estimated by the method described by Decker and Welch [76]. Briefly, to 1 mL of each sample solution prepared at different concentrations (0.1–5 mg/mL), 3.7 mL of distilled water and 100 µL of 2 mM ferrous chloride were added and mixed. Then, the reaction was initiated by the addition of 200 µL of 5 mM ferrozine solution and the mixture was vortexed and kept at room temperature for 10 min. The absorbance of the resulting solution was read at 562 nm (Evolution 201 UV-Visible Spectrophotometer, Thermo Scientific). The control was prepared the same way by using distilled water instead of the sample solution. All determinations were carried out at least in quadruplicate and the EC_50_ value was determined. The percentage of inhibition of ferrozine Fe^2^^+^ complex formation was calculated by the formula:$$\mathrm{Iron}\;\mathrm{chelating}\;\mathrm{activity}\;(\%)=\frac{{\mathrm{Abs}}_{\mathrm{control}}{-\mathrm{Abs}}_{\mathrm{sample}}}{{\mathrm{Abs}}_{\mathrm{control}}}\times 100$$
where Abs_control_ is the absorbance of ferrozine+Fe^2+^complex in the absence of hydrolysate sample and Abs_sample_ is the absorbance of ferrozine+Fe^2+^complex in the presence of hydrolysate sample.

#### 3.6.6. α-Amylase Inhibitory Activity

The α-amylase inhibitory activity was measured using the method described by Hansawasdi et al. [77]. Briefly, the starch azure used as a substrate was previously boiled at a concentration of 1% in 0.05 M Tris-HCl and 0.01 M CaCl_2_ buffer (pH 6.9) for 5 min. After pre-incubation of the starch azure solution at 37 °C for 20 min, 500 μL of this solution were added to 500 μL of hydrolysate sample (25–200 mg/mL) and 500 μL of a porcine pancreatic α-amylase (PPA) solution (2.8 U/mL in the above mentioned buffer). The mixture was incubated for 10 min at 37 °C and the reaction stopped by adding 500 μL of 50% acetic acid. The mixture was then centrifuged at 4500 rpm for 5 min at 4 °C and the absorbance of the resulting supernatant was measured at 595 nm (Evolution 201 UV-Visible Spectrophotometer, Thermo Scientific). The percentage of α-amylase inhibition was calculated as follows:$$\alpha \text{-}\mathrm{Amylase}\;\mathrm{inhibitory}\;\mathrm{activity}\;(\%)=\frac{\left(A_{c+}{-A}_{c-}\right)-\left(A_{s}{-A}_{b}\right)}{A_{c+}{-A}_{c-}}\times 100$$
where A_s_ is the absorbance of the test sample (assay with hydrolysate and PPA), A_b_ is the absorbance of the blank (assay with hydrolysate and without PPA), A_c+_ is the absorbance of the positive control (assay without hydrolysate and with PPA) and A_c−_ is the absorbance of positive control blank (assay without hydrolysate and PPA). All the assays were performed at least in triplicate and the results are presented as mean values.

#### 3.6.7. α-Glucosidase inhibitory activity

The α-glucosidase inhibitory activity was determined according to the methodology described by Kwon et al. [78]. Briefly, 50 µL of FPH solution with different concentrations (25–200 mg/mL) and 100 µL of α-glucosidase (1 U/mL) were pre-incubated at 25 °C for 10 min. After that, 50 µL of the substrate *p*-NPG (5 mM *p*-nitrophenyl–α-D-glucopyranoside in phosphate buffer 0.1 M, pH 6.8) were added. The samples were submitted to an incubation at 25 °C during 5 min. The absorbance was measured at 405 nm in a microplate reader (Bio Rad model 680, Hercules, CA, USA). The percentage of inhibition was expressed by the following equation:$$\alpha \text{-}\mathrm{Glucosidase}\;\mathrm{inhibitory}\;\mathrm{activity}\;(\%)=\frac{\left(A_{c+}{-A}_{c-}\right)-\left(A_{s}{-A}_{b}\right)}{A_{c+}{-A}_{c-}}\times 100$$
where A_s_ is the absorbance of the test sample (assay with hydrolysate and α-glucosidase), A_b_ is the absorbance of the blank (assay with hydrolysate and without α-glucosidase), A_c__+_ is the absorbance of the positive control (assay without hydrolysate and with α-glucosidase) and A_c−_ is the absorbance of positive control blank (assay without hydrolysate and α-glucosidase). All the assays were performed at least in triplicate and the results are presented as mean values. Acarbose (3 mg/mL) was used as a commercial inhibitor and showed an inhibition of 94%.

#### 3.6.8. ACE Inhibitory Activity

The ACE inhibitory activity using Hippuryl-L-Histidyl-L-Leucine (HHL) as substrate was evaluated by high performance liquid chromatography (HPLC) according to the methodology described by Pires et al. [58]. Briefly, 10 μL of FPH solution, 10 μL of borate buffer (used as blank) or captopril 0.0217 mg/mL (standard inhibitor) were mixed with 10 μL of of 0.2 U/mL ACE. The mixture was pre-incubated at 37 °C for 20 min and after this, 50 μL of HHL were added and the mixture was incubated at 37 °C during 30 min. The reaction was stopped by the addition of 85 μL of 1 M HCl, the solution was filtered and an aliquot (10 μL) was injected into a HPLC HP Agilent 1050 series (Agilent, Santa Clara, CA, USA) equipped with a reversed-phase C18 column (100 mm × 4.6 mm, 2.6 m, 100 Å; Kinetex Phenomenex, Alcobendas, Spain). The identity of hippuric acid (HA) and HHL was assessed by comparison with the retention times of standards. The peak areas were obtained with the software Agilent ChemStation for LC (Agilent, Santa Clara, CA, USA) and the percentage of ACE inhibition was calculated as follows:$$\mathrm{ACE}\;\mathrm{inhibitory}\;\mathrm{activity}\;(\%)=\frac{\left({\mathrm{HA}}_{\mathrm{buffer}}{-\mathrm{HA}}_{\mathrm{sample}}\right)}{{\mathrm{HA}}_{\mathrm{buffer}}}\times 100$$
where HA_buffer_ is the concentration of HA in the reaction with the buffer instead of sample and HA_sample_ is the concentration of HA in the reaction with the sample. Captopril was used as a commercial inhibitor and showed an ACE inhibitory activity of ca. 90–100%.

### 3.7. Statistical Analysis

All statistical analyses were performed using the software STATISTICA© version 12 (data analysis software system) from StatSoft, Inc. (Tulsa, OK, USA). The results of the analyses are reported as mean values ± standard deviation (SD) and differences between mean values were performed using a one-way analysis of variance (ANOVA). For this, the Tukey test was applied with a significance value of *p* < 0.05.

A multivariate analysis of the relationship between amino acids composition, antioxidant activities (DPPH, ABTS, Reducing Power), chelating activities (Fe^2+^ and Cu^2+^), molecular weight profile, anti-diabetic and anti-hypertensive activities was performed by Principal Components Analysis and the two main factors were represented.

## 4. Conclusions

The hydrolysates had high protein content and contained all the essential amino acids, which demonstrated their potential use as supplements for human nutrition.

All FPH exhibited antioxidant activity, which depended on the species and the part of the fish used in the hydrolysates’ preparation. Among hydrolysates, AHM_He and Gu_He presented the highest antioxidant activity.

Blue whiting hydrolysates exhibited the highest α-amylase inhibitory activity and RS_Sb showed the highest α-glucosidase inhibitory activity. All hydrolysates showed high percentage of ACE inhibition for the 5 mg/mL hydrolysate, especially M_Wh. The biological activities of FPH make them a potential natural additive for functional foods or nutraceuticals. However, further work is necessary to check these activities in vivo and isolate and identify the peptides responsible for those biological activities.

## Figures and Tables

**Figure 1 marinedrugs-19-00338-f001:**
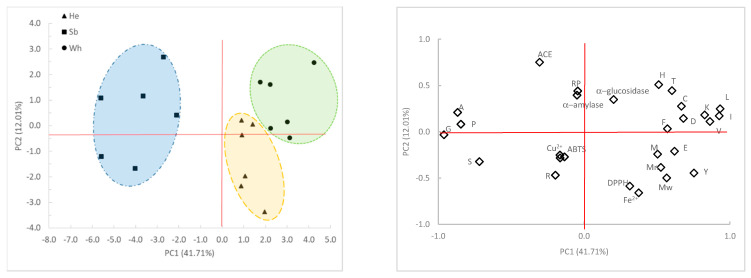
Principal component analysis (PCA) of amino acid composition, antioxidant activities (DPPH, ABTS, RP), chelating activities (Fe^2+^ and Cu^2+^), molecular weight profile (Mn and Mw), α-amylase, α-glucosidase and ACE inhibitory activities of FPH prepared from heads (He), skins and bones (Sb) and whole fish (Wh) of fish discards. A-Ala, R-Arg, D-Asp, C-Cys, E-Glu, G-Gly, H-His, I-Ile, L-Leu, K-Lys, M-Met, F-Phe, P-Pro, S-Ser, T-Thr, Y-Tyr, V-Val.

**Table 1 marinedrugs-19-00338-t001:** Yield, chemical composition and molecular weight of dried FPH prepared with heads (He), skins and bones (Sb) and whole fish (Wh) of fish discards. Y: yield of FPH production, Mo: moisture, OM: organic matter, TP: total protein, TF: total fat, Dig: digestibility, Mn: number average molecular weight, Mw: average molecular weight. BW: blue whiting, RS: red scorpionfish, Po: pouting, Gu: gurnard, M: megrim and AHM: Atlantic horse mackerel.

FPH	Y (%)	Mo (%)	OM (%)	Ash (%)	TP (%)	TF (%)	Dig (%)	Mn (Da)	Mw (Da)
BW_Sb	97.3 ± 1.2	5.0 ± 0.2 ^a,b,c^	79.6 ± 0.3	15.4 ± 0.6 ^d^	73.5 ± 2.9	5.0 ± 2.4 ^a,b,c,d,e^	93.4 ± 0.4 ^c,d,e,f,g^	632 ± 26 ^a,^^b^	940 ± 49 ^a^
RS_Sb	93.2 ± 0.8	3.5 ± 0.1 ^a,b^	82.1 ± 3.4	14.4 ± 3.4 ^b,c,d^	74.7 ± 2.6	6.2 ± 1.8 ^a,b,c,d,e^	94.4 ± 1.0 ^d,e,f,g,h^	557 ± 38 ^a^	848 ± 80 ^a^
Po_Sb	95.7 ± 2.8	4.1 ± 0.9 ^a,b,c^	83.1 ± 0.7	12.9 ± 0.1 ^a,b,c^	76.6 ± 3.0	4.8 ± 0.3 ^a,b,c,d,e^	95.8 ± 1.1 ^g,h^	634 ± 47 ^a,b^	954 ± 75 ^a^
Gu_Sb	94.7 ± 2.6	6.7 ± 2.1 ^c,d^	81.5 ± 2.2	11.9 ± 0.2 ^a^	74.5 ± 1.9	4.4 ± 0.6 ^a,b,c,d,e^	92.7 ± 1.1 ^b,c,d,e^	631 ± 100 ^a,b^	941 ± 107 ^a^
M_Sb	90.2 ± 1.9	3.3 ± 1.1 ^a^	82.7 ± 1.7	14.0 ± 0.6 ^a,b,c,d^	75.2 ± 4.0	3.6 ± 0.1 ^a,b,c,d^	93.9 ± 0.8 ^d,e,f,g^	733 ± 76 ^a,b,c^	1168 ± 133 ^a,b,c^
AHM_Sb	88.3 ± 1.4	5.0 ± 2.5 ^a,b,c^	83.0 ± 2.6	12.1 ± 0.1 ^a,b^	72.1 ± 4.1	7.9 ± 1.1 ^c,d,e^	92.8 ± 0.5 ^b,c,d,e,f^	1016 ± 141 ^c,d,e^	1890 ± 113 ^e^
BW_He	87.7 ± 1.7	5.1 ± 0.7 ^a,b,c^	81.0 ± 0.7	14.0 ± 0.2 ^a,b,c,d^	70.1 ± 3.0	9.4 ± 5.8 ^e^	92.3 ± 0.5 ^a,b,c,d,e^	1081 ± 78 ^d,e^	2353 ± 98 ^f^
RS_He	87.5 ± 1.5	5.2 ± 1.2 ^a,b,c,d^	81.8 ± 2.3	13.0 ± 1.1 ^a,b,c,d^	70.2 ± 3.1	8.0 ± 5.1 ^d,e^	89.7 ± 0.9 ^a^	1099 ± 125 ^d,e^	2490 ± 181 ^f,g^
Po_He	90.4 ± 0.8	4.9 ± 0.3 ^a,b,c^	80.5 ± 0.2	14.7 ± 0.5 ^c,d^	71.1 ± 4.1	2.3 ± 0.7 ^a,b,c^	91.7 ± 0.7 ^a,b,c,d^	992 ± 84 ^c,d,e^	2329 ± 191 ^f^
Gu_He	85.3 ± 1.2	5.6 ± 0.3 ^a,b,c,d^	80.4 ± 1.5	14.1 ± 1.2 ^a,b,c,d^	71.9 ± 4.8	3.3 ± 0.4 ^a,b,c,d^	90.7 ± 0.3 ^a,b,c^	1247 ± 134 ^e^	2823 ± 122 ^g^
M_He	86.1 ± 0.7	6.3 ± 0.4 ^b,c,d^	79.7 ± 0.6	14.1 ± 0.2 ^a,b,c,d^	69.8 ± 2.0	2.5 ± 0.2 ^a,b,c,d^	92.0 ± 0.1 ^a,b,c,d,e^	1785 ± 127 ^f^	2875 ± 225 ^g^
AHM_He	90.3 ± 2.2	5.1 ± 1.0 ^a,b,c^	81.1 ± 1.2	13.9 ± 0.2 ^a,b,c,d^	70.4 ± 1.8	7.2 ± 1.7 ^b,c,d,e^	90.3 ± 0.5 ^a,b^	1203 ± 136 ^e^	2359 ± 237 ^f^
BW_Wh	93.1 ± 0.5	5. 2 ± 0.8 ^a,b,c,d^	81.2 ± 0.4	13.6 ± 0.4 ^a,b,c,d^	74.7 ± 2.8	3.8 ± 0.5 ^a,b,c,d^	97.2 ± 0.4 ^h^	779 ± 117 ^a,b,c^	1428 ± 137 ^c,d^
RS_Wh	88.5 ± 0.9	5.5 ± 1.0 ^a,b,c,d^	81.7 ± 0.9	12.8 ± 0.0 ^a,b,c^	75.8 ± 1.9	1.5 ± 0.7 ^a^	94.6 ± 2.5 ^e,f,g,h^	823 ± 117 ^a,b,c,d^	1394 ± 173 ^b,c,d^
Po_Wh	91.6 ± 0.7	8.0 ± 1.4 ^d^	79.3 ± 2.2	12.7 ± 0.8 ^a,b,c^	73.0 ± 3.8	2.9 ± 0.3 ^a,b,c,d^	93.2 ± 0.4 ^b,c,d,e,f,g^	651 ± 100 ^a,b^	978 ± 126 ^a,b^
Gu_Wh	83.4 ± 1.1	5.1 ± 1.5 ^a,b,c^	80.9 ± 1.6	14.0 ± 0.1 ^a,b,c,d^	75.8 ± 3.2	1.7 ± 0.6 ^a,b^	94.4 ± 0.7 ^d,e,f,g,h^	1016 ± 85 ^c,d,e^	1809 ± 85 ^d,e^
M_Wh	90.9 ± 3.1	5.0 ± 0.5 ^a,b,c^	81.7 ± 0.5	13.3 ± 0.0 ^a,b,c,d^	74.4 ± 0.2	4.2 ± 0.1 ^a,b,c,d,e^	95.7 ± 0.9 ^f,g,h^	1232 ± 96 ^e^	2350 ± 177 ^f^
AHM_Wh	90.0 ± 2.1	6.4 ± 0.1 ^c,d^	79.8 ± 0.4	13.8 ± 0.5 ^a,b,c,d^	71.9 ± 1.8	3.0 ± 2.5 ^a,b,c,d^	94.3 ± 2.8 ^d,e,f.g.h^	874 ± 120 ^b,c,d^	1669 ± 130 ^d,e^

Results are expressed as mean ± standard deviations. Different letters in same column indicate significantly different means (*p* < 0.05).

**Table 2 marinedrugs-19-00338-t002:** Non-essential amino acids content of FPH (% or g/100 g total amino acids) from head (He), skins and bones (Sb) and whole fish (Wh) of fish discards. BW: blue whiting, RS: red scorpionfish, Po: pouting, Gu: gurnard, M: megrim and AHM: Atlantic horse mackerel. Errors are the confidence intervals for *n* = 2 and α = 0.05.

	NEAA (%)									
FPH	Asp	Ser	Glu	Ala	Gly	Cys	Tyr	Arg	Pro	Hyp
BW_Sb	9.85 ± 0.071 ^c,d,e^	5.38 ± 0.184 ^b,c,d,e^	15.01 ± 0.438 ^c,d,e^	6.35 ± 0.007 ^a^	7.18 ± 0.248 ^a,b^	0.39 ± 0.064 ^a,b,c^	3.72 ± 0.212 ^d,e,f^	7.01 ± 0.106 ^c^	4.37 ± 0.438 ^a,b,c^	4.14 ± 0.304 ^c,d^
RS_Sb	9.84 ± 0.233 ^c,d,e^	5.12 ± 0.262 ^b,c,d^	15.32 ± 0.410 ^c,d,e^	6.23 ± 0.226 ^a^	6.99 ± 0.346 ^a,b^	0.41 ± 0.042 ^a,b,c,d^	4.13 ± 0.283 ^f,g^	6.78 ± 0.856 ^b,c^	4.47 ± 0.587 ^a,b,c^	3.98 ± 0.156 ^c,d^
Po_Sb	10.13 ± 0.028 ^d^,e	5.12 ± 0.049 ^b,c,d^	13.83 ± 0.057 ^a,b,c,d^	7.68 ± 0.035 ^c,d^	8.29 ± 0.113 ^b^	0.36 ± 0.007 ^a,b^	3.13 ± 0 ^b,c,d,e^	6.095 ± 0.021 ^a,b,c^	4.22 ± 0.014 ^a,b^	4.42 ± 0.283 ^d^
Gu_Sb	9.95 ± 0.057 ^c,d,e^	5.17 ± 0.354 ^b,c,d,e^	15.34 ± 1.789 ^c,d,e^	6.64 ± 0.481 ^a,b^	7.44 ± 1.322 ^a,b^	0.39 ± 0.007 ^a,b,c^	3.42 ± 0.078 ^b,c,d,e,f^	6.23 ± 0.141 ^a,b,c^	4.92 ± 0.014 ^b,c,d,e^	4.08 ± 0.304 ^c,d^
M_Sb	9.88 ± 0.141 ^c,d,e^	5.26 ± 0.474 ^b,c,d,e^	16.33 ± 0.481 ^e^	6.50 ± 0.290 ^a,b^	6.73 ± 0.325 ^a,b^	0.36 ± 0.028 ^a,b^	3.77 ± 0.417 ^e,f^	6.31 ± 0.191 ^a,b,c^	4.89 ± 0.021 ^b,c,d,e^	4.28 ± 0.028 ^c,d^
AHM_Sb	9.45 ± 0.233 ^b,c^	5.34 ± 0.106 ^b,c,d,e^	14.12 ± 0.276 ^a,b,c,d^	6.57±0.035 ^a,b^	6.73 ± 0.163 ^a,b^	0.45 ± 0.000 ^b,c,d^	4.78 ± 0.566 ^g^	6.76 ± 0.226 ^b,c^	4.62 ± 0.092 ^a,b,c,d^	3.50 ± 0.304 ^a,b,c,d^
BW_He	9.77 ± 0.035 ^b,c,d,e^	5.81 ± 0.071 ^e^	14.45 ± 0.170 ^b,c,d,e^	8.25 ± 0.240 ^d,e,f^	10.88 ± 0.219 ^c^	0.35 ± 0.021 ^a,b^	2.87 ± 0.021 ^a,b,c^	6.25 ± 0.035 ^a,b,c^	5.43 ± 0.007 ^c,d,e^	1.99 ± 0.226 ^a^
RS_He	9.13 ± 0.021 ^b^	5.38 ± 0.078 ^b,c,d,e^	12.88 ± 0.191 ^a,b^	8.21 ± 0.035 ^d,e,f^	11.64 ± 0.049 ^c^	0.27 ± 0.007 ^a^	2.81 ± 0.035 ^a,b,c^	6.52 ± 0.078 ^b,c^	6.45 ± 0.148 ^f^	2.21 ± 0.198 ^a,b^
Po_He	9.64 ± 0.035 ^b,c,d^	5.50 ± 0.064 ^c,d,e^	13.50 ± 0.007 ^a,b,c^	8.47 ± 0.134 ^d,e,f^	12.36 ± 0.255 ^c^	0.32 ± 0.007 ^a,b^	2.76 ± 0.007 ^a,b,c^	6.63 ± 0.064 ^b,c^	6.31 ± 0.057 ^f^	2.78 ± 0.113 ^a,b,c^
Gu_He	9.61 ± 0.057 ^b,c,d^	5.15 ± 0.007 ^b,c,d,e^	12.55 ± 0.127 ^a,b^	8.10 ± 0.035 ^d,e^	11.46 ± 0.113 ^c^	0.33 ± 0.000 ^a,b^	2.64 ± 0.056 ^a,b^	5.85 ± 0.127 ^a,b^	5.11 ± 0.212 ^b,c,d,e^	9.41 ± 0.926 ^e^
M_He	9.12 ± 0.361 ^b^	5.86 ± 0.078 ^e^	12.49 ± 0.007 ^a,b^	8.85 ± 0.071 ^e,f^	15.05 ± 0.057 ^d^	0.26 ± 0.014 ^a^	2.29 ± 0.021 ^a^	6.66 ± 0.304 ^b,c^	6.24 ± 0.34 ^f^	3.24 ± 0.276 ^a,b,c,d^
AHM_He	7.98 ± 0.156 ^a^	5.85 ± 0.049 ^e^	12.36 ± 0.240 ^a^	9.01 ± 0.106 ^f^	15.94 ± 0.233 ^d^	0.29 ± 0.042 ^a^	2.81 ± 0.085 ^a,b,c^	6.47 ± 0.198 ^b,c^	5.93 ± 0.099 ^e,f^	3.60 ± 0.304 ^b,c,d^
BW_Wh	10.39 ± 0.064 ^e^	5.13 ± 0.092 ^b,c,d^	14.9 ± 0.198 ^c,d,e^	7.26 ± 0.099 ^b,c^	5.96 ± 0.127 ^a^	0.54 ± 0.014 ^d,e^	3.57 ± 0.113 ^c,d,e,f^	6.00 ± 0.057 ^a,b^	3.72 ± 0.205 ^a^	3.04 ± 0.375 ^a,b,c,d^
RS_Wh	9.86 ± 0.071 ^c,d,e^	4.82 ± 0.127 ^a,b,c^	13.39 ± 0.354 ^a,b,c^	6.98 ± 0.035 ^a,b,c^	5.78 ± 0.106 ^a^	0.61 ± 0.028 ^f^	3.47 ± 0.042 ^c,d,e,f^	5.41 ± 0.071 ^a^	3.68 ± 0.156 ^a^	3.14 ± 0.184 ^a,b,c,d^
Po_Wh	10.23 ± 0.049 ^d,e^	5.02 ± 0.014 ^b,c^	15.81 ± 0.268 ^d,e^	7.25 ± 0.042 ^b,c^	5.92 ± 0.035 ^a^	0.41 ± 0.014 ^a,b,c,d^	2.94 ± 0.021 ^a,b,c,d^	6.37 ± 0.042 ^a,b,c^	4.48 ± 0.028 ^a,b,c^	2.94 ± 0.212 ^a,b,c,d^
Gu_Wh	10.02 ± 0.014 ^c,d,e^	4.77 ± 0.042 ^a,b^	13.87 ± 0.071 ^a,b,c,d^	7.17 ± 0.085 ^b,c^	6.17 ± 0.014 ^a^	0.55 ± 0.021 ^d,e^	3.56 ± 0.049 ^c,d,e,f^	6.14 ± 0.057 ^a,b,c^	4.07 ± 0.191 ^a,b^	3.305 ± 0.205 ^kb,c,d^
M_Wh	9.38 ± 0.375 ^b,c^	4.31 ± 0.134 ^a^	13.74 ± 0.375 ^a,b,c^	6.96 ± 0.247 ^a,b,c^	8.63 ± 1.386 ^a^	0.85 ± 0.120 ^e^	3.42 ± 0.297 ^b,c,d,e,f^	6.71 ± 0.233 ^b,c^	5.63 ± 0.559 ^d,e,f^	2.97 ± 0.148 ^a,b,c,d^
AHM_Wh	10.01 ± 0.262 ^c,d,e^	4.955 ± 0.177 ^a,b,c^	13.84 ± 0.368 ^a,b,c,d^	6.92 ± 0.424 ^a,b,c^	6.27 ± 0.078 ^a^	0.53 ± 0.028 ^c,d,e^	3.38 ± 0.035 ^b,c,d,e,f^	6.29 ± 0.163 ^a,b,c^	4.19 ± 0.304 ^a,b^	3.68 ± 0.891 ^b,c,d^

Results are expressed as mean ± standard deviations. Different letters in same column indicate significantly different means (*p* < 0.05).

**Table 3 marinedrugs-19-00338-t003:** Essential amino acids content of FPH (% or g/100 g total amino acids) prepared with heads (He), skins and bones (Sb) and whole fish (Wh) of fish discards. TEAA/TAA: total essential amino acids per total amino acids. BW: blue whiting, RS: red scorpionfish, Po: pouting, Gu: gurnard, M: megrim and AHM: Atlantic horse mackerel. Errors are the confidence intervals for n = 2 and α = 0.05.

FPH	EAA (%)								
	Thr	Val	Met	Ile	Leu	Phe	His	Lys	TEAA/TAA
BW_Sb	4.16 ± 0.099 ^a,b,c^	4.20 ± 0.028 ^d,e^	3.56 ± 0.057 ^c,d^	3.37 ± 0.537 ^a.b.c.d^	7.10 ± 0.079 ^b,c^	4.16 ± 0.127 ^a,b,c^	2.22 ± 0.297 ^a,b,c^	7.82 ± 0.042 ^f,g,h^	36.59 ± 1.068
RS_Sb	4.02 ± 0.283 ^a,b,c^	4.23 ± 0.113 ^d,e^	3.69 ± 0.127 ^c,d^	3.40 ± 0.417 ^a,b,c,d^	7.14 ± 0.049 ^b,c,d,e^	4.49 ± 0.587 ^b,c^	2.14 ± 0.495 ^a,b,c^	7.59 ± 0.071 ^e,f,g^	36.69 ± 0.742
Po_Sb	4.27 ± 0.042 ^b,c,d^	4.21 ± 0.021 ^d,e^	3.66 ± 0.014 ^c,d^	3.33 ± 0.021 ^a,b,c,d^	7.41 ± 0.021 ^c,d,e,f^	4.79 ± 0.028 ^c^	1.95 ± 0.007 ^a,b^	7.09 ± 0.042 ^c,d,e,f^	36.69 ^e^ ± 0.156
Gu_Sb	4.37 ± 0.163 ^b,c,d^	4.21 ± 0.283 ^d,e^	3.27 ± 0.085 ^b,c,d^	3.94 ± 0.764 ^c,d^	7.59 ± 0.205 ^d,e,f,g^	3.86 ± 0.495 ^a,b,c^	2.07 ± 0.403 ^a,b,c^	7.04 ± 0.325 ^b,c,d,e^	36.335 ± 0.389
M_Sb	4.29 ± 0.042 ^b,c,d^	4.12 ± 0.141 ^d,e^	3.47 ± 0.361 ^c,d^	3.80 ± 0.969 ^c,d^	7.52 ± 0.028 ^c,d,ef,g^	3.26 ± 0.361 ^a^	2.23 ± 0.127 ^a,b,c^	7.03 ± 0.311 ^b,c,d,e^	35.71 ± 0.318
AHM_Sb	4.29 ± 0.057 ^b,c,d^	4.39 ± 0.071 ^d,e^	3.93 ± 0.460 ^d^	3.53 ± 0.255 ^b,c,d^	7.12 ± 0.247 ^b,c,d^	4.38 ± 0.226 ^b,c^	2.57 ± 0.049 ^b,c,d,e^	7.48 ± 0.368 ^d,e,f^	37.68 ± 0.813
BW_He	4.45 ± 0.134 ^b,c,d^	3.88 ± 0.071 ^b,c,d,e^	3.22 ± 0.255 ^b,c,d^	2.96 ± 0.085 ^a,b,c,d^	6.76 ± 0.035 ^b^	3.97 ± 0.120 ^a,b,c^	1.95 ± 0.000 ^a,b^	6.77 ± 0.092 ^a,b,c,d^	33.94 ± 0.368
RS_He	4.36 ± 0.057 ^b,c,d^	4.03 ± 0.014 ^c,d,e^	3.05 ± 0.078 ^b,c^	3.13 ± 0.014 ^a,b,c,d^	6.83 ± 0.049 ^b^	3.86 ± 0.134 ^a,b,c^	2.72 ± 0.184 ^c,d,e^	6.56 ± 0.050 ^a,b,c^	34.52 ± 0.184
Po_He	4.18 ± 0.042 ^a,b,c^	3.27 ± 0.000 ^a,b^	3.24 ± 0.092 ^b,c,d^	2.42 ± 0.035 ^a,b,c^	6.13 ± 0.064 ^a^	3.98 ± 0.021 ^a,b,c^	1.80 ± 0.000 ^a^	6.73 ± 0.042 ^a,b,c,d^	31.73 ± 0.226
Gu_He	3.47 ± 0.078 ^a^	2.93 ± 0.035 ^a^	3.31 ± 0.085 ^b,c,d^	2.06 ± 0.049 ^a,b^	5.87 ± 0.042 ^a^	4.00 ± 0.021 ^a,b,c^	1.93 ± 0.014 ^a,b^	6.21 ± 0.240 ^a,b^	29.76 ± 0.410
M_He	4.02 ± 0.078 ^a,b,c^	3.30 ± 0.042 ^a,b,c^	3.30 ± 0.191 ^b,c,d^	2.06 ± 0.120 ^a,b^	5.72 ± 0.071 ^a^	3.70 ± 0.184 ^a,b^	1.91 ± 0.035 ^a,b^	5.93 ± 0.233 ^a^	29.92 ± 0.728
AHM_He	4.0 ± 0.014 ^a,b^	3.67 ± 0.269 ^b,c,d^	2.12 ± 0.057 ^a^	1.99 ± 0.092 ^a^	5.68 ± 0.064 ^a^	4.05 ± 0.290 ^a,b,c^	2.29 ± 0.177 ^a,b,c,d^	5.94 ± 0.346 ^a^	29.73 ± 0.587
BW_Wh	4.40 ± 0.113 ^b,c,d^	4.52 ± 0.014 ^e,f^	3.63 ± 0.106 ^c,d^	3.72 ± 0.049 ^c,d^	8.36 ± 0.049 ^h^	4.79 ± 0.134 ^c^	2.02 ± 0.049 ^a,b,c^	8.52 ± 0.078 ^h,i^	39.93 ± 0.226
RS_Wh	4.73 ± 0.092 ^c,d^	5.25 ± 0.049 ^f^	3.57 ± 0.120 ^c,d^	4.32 ± 0.028 ^d^	8.34 ± 0.049 ^h^	4.58 ± 0.163 ^b,c^	4.34 ± 0.092 ^f^	7.72 ± 0.092 ^f,g,h^	42.82 ± 0.021
Po_Wh	4.99 ± 0.049 ^d^	4.00 ± 0.057 ^b,c,d,e^	2.61 ± 0.049 ^a,b^	3.71 ± 0.078 ^c,d^	7.89 ± 0.042 ^g,h^	4.12 ± 0.042 ^a,b,c^	2.62 ± 0.01 ^b,c,d,e^	8.86 ± 0.035 ^i^	38.78 ± 0.184
Gu_Wh	4.45 ± 0.120 ^b,c,d^	4.50 ± 0.078 ^e^	3.54 ± 0.134 ^c,d^	3.75 ± 0.099 ^c,d^	7.97 ± 0.021 ^g,h^	4.66 ± 0.106 ^b,c^	3.23 ± 0.071 ^e^	8.39 ± 0.057 ^g,h,i^	40.465 ± 0.686
M_Wh	4.62 ± 0.141 ^b,c,d^	4.16 ± 0.071 ^d,e^	3.06 ± 0.184 ^b,c^	4.01 ± 0.113 ^d^	7.60 ± 0.276 ^e,f,g^	4.55 ± 0.163 ^b,c^	2.39 ± 0.212 ^a,b,c,d^	7.19 ± 0.445 ^c,d,e,f^	37.57 ± 0.346
AHM_Wh	4.49 ± 0.594 ^b,c,d^	4.22 ± 0.629 ^d,e^	3.21 ± 0.403 ^b,c,d^	3.45 ± 0.707 ^a,b,c,d^	7.65 ± 0.198 ^f,g^	4.49 ± 0.269 ^b,c^	2.99 ± 0.014 ^d,e^	8.68 ± 0.113 ^i^	39.17 ± 0.608

Results are expressed as mean ± standard deviations. Different letters in same column indicate significantly different means (*p* < 0.05).

**Table 4 marinedrugs-19-00338-t004:** Antioxidant (DPPH and ABTS radical scavenging activities and reducing power) and chelating activities (Cu^2+^ and Fe^2+^) prepared with heads (He), skins and bones (Sb) and whole fish (Wh) of fish discards. BW: blue whiting, M: megrim, RS: red scorpionfish, Po: pouting, Gu: gurnard, and AHM: Atlantic horse mackerel.

FPH	DPPH	ABTS	Reducing Power	Cu^2+^	Fe^2+^
	% Inhibition (3 mg/mL)	EC_50_ (mg/mL)	A_0.5_ (mg/mL)	EC_50_ (mg/mL)	
BW_He	18.91 ± 0.119 ^h^	4.00 ± 0.057 ^h^	3.39 ± 0.034 ^b^	5.66 ± 0.095 ^h^	0.38 ± 0.009 ^d,e,f^
M_He	4.46 ± 0.536 ^b^	2.17 ± 0.055 ^b^	6.35 ± 0.037 ^m^	3.13 ± 0.095 ^b,c,d,e^	0.35 ± 0.023 ^b,c,d,e,f^
RS_He	31.87 ± 0.655 ^j^	2.45 ± 0.059 ^b,c,d^	4.58 ± 0.131 ^j^	3.73 ± 0.057 ^f^	0.49 ± 0.095 ^g,h^
Po_He	14.04 ± 0.335 ^g^	4.01 ± 0.066 ^h^	3.98 ± 0.013 ^f,g^	3.47 ± 0.061 ^d,e,f^	0.32 ± 0.040 ^a,b,c,d,e^
Gu_He	8.65 ± 0.549 ^d,e^	1.12 ± 0.026 ^a^	3.54 ± 0.029 ^b,c^	3.41 ± 0.025 ^d,e,f^	0.26 ± 0.002 ^a^
AHM_He	29.22 ± 0.067 ^j^	4.89 ± 0.091 ^i^	3.19 ± 0.057 ^a^	2.49 ± 0.019 ^a^	0.53 ± 0.012 ^h^
BW_Sb	2.83 ± 0.293 ^a,b^	4.76 ± 0.571 ^i^	4.07 ± 0.011 ^g,h^	5.20 ± 0.051 ^h^	0.28 ± 0.011 ^a,b^
M_Sb	NA	2.95 ± 0.132 ^e,f^	5.79 ± 0.031 ^l^	3.40 ± 0.060 ^c,d,e,f^	0.26 ± 0.005 ^a^
RS_Sb	NA	2.72 ± 0.079 ^d,e^	4.57 ± 0.022 ^j^	3.32 ± 0.021 ^c,d,e,f^	0.28 ± 0.004 ^a,b^
Po_Sb	14.75 ± 0.170 ^g^	3.68 ± 0.008 ^g,h^	4.22 ± 0.149 ^h,i^	3.34 ± 0.021 ^c,d,e,f^	0.27 ± 0.004 ^a^
Gu_Sb	0.76 ± 0.340 ^a^	2.22 ± 0.073 ^b,c^	4.37 ± 0.003 ^i^	3.66 ± 0.025 ^e,f^	0.31 ± 0.019 ^a,b,c,d^
AHM_Sb	25.10 ± 0.829 ^i^	4.93 ± 0.018 ^i^	3.51 ± 0.009 ^b,c^	3.22 ± 0.031 ^c,d,e,f^	0.40 ± 0.002 ^e,f^
BW_Wh	10.80 ± 1.041 ^e,f^	3.30 ± 0.037 ^f,g^	3.56 ± 0.014 ^b,c^	4.67 ± 0.031 ^g^	0.36 ± 0.011 ^a,d,e,f^
M_Wh	12.29 ± 0.382 ^c,d^	2.67 ± 0.131 ^c,d,e^	5.10 ± 0.010 ^k^	3.07 ± 0.028 ^b,c,d^	0.27 ± 0.008 ^a,b^
RS_Wh	12.29 ± 0.382 ^f,g^	2.24 ± 0.094 ^b,c^	4.97 ± 0.010 ^k^	2.75 ± 0.022 ^a,b,c^	0.43 ± 0.015 ^f,g^
Po_Wh	18.26 ± 0.130 ^h^	3.81 ± 0.024 ^h^	3.84 ± 0.049 ^e,f^	3.14 ± 0.035 ^b,c,d,e,f^	0.31 ± 0.017 ^a,b,c,d^
Gu_Wh	5.22 ± 1.147 ^b,c^	1.47 ± 0.020 ^a^	3.65 ± 0.018 ^c,d^	3.37 ± 0.037 ^c,d,e,f^	0.38 ± 0.014 ^c,d,e,f^
AHM_Wh	18.44 ± 0.850 ^h^	4.56 ± 0.039 ^i^	3.75 ± 0.011 ^d,e^	2.67 ± 0.023 ^a,b^	0.29 ± 0.010 ^a,b,c^

Results are expressed as mean ± standard deviations. Different letters in same column indicate significantly different means (*p* < 0.05).

**Table 5 marinedrugs-19-00338-t005:** ACE, α-amylase and α-glucosidase inhibitory activities of FPH prepared with heads (He), skins and bones (Sb) and whole fish (Wh) of fish discards. BW: blue whiting, M: megrim, RS: red scorpionfish, Po: pouting, Gu: gurnard and AHM: Atlantic horse mackerel.

FPH	IC_50_ (mg/mL)		% Inhibition (5 mg/mL)
	α-Amylase	α-Glucosidase	ACE
BW_He	5.70 ± 0.67 ^a^	---	61.20 ± 7.83 ^a^
M_He	18.49 ± 1.90 ^b,c,d^	216.9 ± 26.3 ^g,h,i^	76.58 ± 2.72 ^a,b,c^
RS_He	---	155.5 ± 4.1 ^d,e,f,g^	64.22 ± 2.46 ^a,b^
Po_He	60.29 ± 9.46 ^g^	154.4 ± 3.5 ^d,e,f,g^	79.35 ± 17.01 ^a,b,c^
Gu_He	17.42 ± 0.49 ^a,b,c,d^	247.8 ± 19.9 ^h,i,j^	69.28 ± 5.10 ^a,b,c^
AHM_He	24.96 ± 6.73 ^c,d,e^	83.1 ± 4.0 ^b,c^	60.77 ± 2.44 ^a^
BW_Sb	13.96 ± 0.71 ^a,b^	---	77.68 ± 2.15 ^a,b,c^
M_Sb	33.66 ± 2.93 ^e,f^	103.7 ± 15.2 ^b,c,d^	79.40 ± 2.01 ^a,b,c^
RS_Sb	27.14 ± 0.96 ^c,d,e^	21.8 ± 8.9 ^a^	77.79 ± 1.06 ^a,b,c^
Po_Sb	33.27 ± 1.04 ^e^	133.7 ± 2.9 ^a,c,d,e^	82.10 ± 2.72 ^b,c^
Gu_Sb	27.82 ± 9.75 ^c,d,e^	214.5 ± 9.0 ^g,h,i^	82.41 ± 2.19 ^b,c^
AHM_Sb	29.33 ± 2.34 ^d,e^	75.1 ± 1.5 ^b^	70.71 ± 3.09 ^a,b,c^
BW_Wh	18.51 ± 1.53 ^b,c,d^	156.9 ± 7.2 ^d,e,f,g^	82.69 ± 12.81 ^b,c^
M_Wh	44.06 ± 2.65 ^f^	300.0 ± 25.8 ^j^	85.95 ± 0.56 ^c^
RS_Wh	23.50 ± 1.49 ^b,c,d,e^	256.7 ± 58.1 ^i,j^	77.07 ± 1.02 ^a,b,c^
Po_Wh	---	187.7 ± 36.1 ^e,f,g,h^	66.60 ± 6.22 ^a,b,c^
Gu_Wh	16.71 ± 0.25 ^a,b,c^	203.5 ± 14.8 ^f,g,h,i^	72.78 ± 5.57 ^a,b,c^
AHM_Wh	84.37 ± 5.21 ^h^	143.3 ± 0.8 ^c,d,e,f^	75.58 ± 4.48 ^a,b,c^

Results are expressed as mean ± standard deviations. Different letters in same column indicate significantly different means (*p* < 0.05).

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

*Melongena*) phenolics as inhibitors of key enzymes relelvant for type 2 diabetes and hypertension. Bioresour. Technol..

