# Peer review of "Characterization of Protein Hydrolysates from Fish Discards and By-Products from the North-West Spain Fishing Fleet as Potential Sources of Bioactive Peptides"

_marinedrugs, 2021, doi:10.3390/md19060338_

Round 1

Reviewer 1 Report

Marine organisms is a source of important ingredients in the preparation of medicinal, protective and prophylactic medicines. Medicines of animal origin are part of traditional medicine in different countries, including China, Russia, India, and others. While reading the manuscript, I had some questions and recommendations. 

  1. Medicines from production marine organisms are widely used in medicine (https://doi.org/10.1016/j.jep.2019.111933). Please discuss these issues in more detail.
  2. The questions of utilization of fish peptides were previously considered and discussed in the article (https://doi.org/10.1134/S1063074012060090).
  3. The bioavailability of peptides including the source additive for functional foods or nutraceuticals is an extremely important aspect. Please discuss the bioavailability and routes of administration of peptides (https://doi.org/10.3390/md18110557).
  4. The FPHs were high in fat. Has the storage stability of hydrolysates been studied? Please provide data on at least short-term storage under stressful conditions.
  5. In table 1, indicate the yield of hydrolysates.

Author Response

Dear Reviewer,

Thank you very much for your comments about our manuscript which were very useful and allowed to improve it. 

Comments 1-3

Thank you very much for your recommendations about the discussion on pharmaceutical products from marine organisms and bioavailability of peptides as a source of nutraceuticals or ingredients for functional foods. We include this issues on the introduction of manuscript. We also considered the article about the biological activities of fish peptides and methods of their isolation.

Comment 4

Storage stability was not studied because the FPH samples were immediately vacuum-packed after production and drying, and stored at -18ºC until use. We have included these details in the new version of the manuscript:

“To maintain their stability, dry FPH were subsequently vacuum-packed and stored at -18ºC until use.”

In addition, the objective of the study was to compare the biological activities of hydrolysates prepared in the same conditions from various fish discards and by-products. So, we did not consider the study of storage stability. 

Comment 5

We have included the yield of hydrolysates in Table 1.

Reviewer 2 Report

This article presents interesting results concerning the utilization of fishery byproducts to produce fish protein hydrolysates (FPHs) via the enzymatic method and bioactivity evaluation of FPHs (such as antioxidant, anti-diabetic, and ACE inhibitory activity). Here are some critical points:

- Line 41-42: “These by-products are usually composed of heads (9-12% of total fish weight), viscera (12-18%), skin (1-3%), bones (9-15%) and scales (about 5%)”. This sentence indicates that viscera is the major by-product of fish. Why did the authors not use this kind of material but only the head and skin of the fish? The selection of material sources for hydrolysis should therefore be conducted more logically.

- Line 62-63: This sentence should be linked to the next paragraph.

- The reason for choosing Alcalase for fish protein hydrolysis should also be mentioned.

- The subsubsections of 2.6, 2.7, 2.8, etc. subsections should be numbered (2.6.1, 2.6.2, etc.)

- Section 3.1: How to confirm that the amino acids and peptides are products of the enzymatic hydrolysis? Therefore, the authors are suggested to add the results of chemical composition and other characteristics (Mw, moisture, etc.) of the original fish materials.  Furthermore, a comparison should be made between the materials and the products after the hydrolysis has taken place.

- Table 4. Are EC50 and A0.5 similar?. Also, at line 55 and 56, the authors used AC0.5 and A0.5. Please consider them. Why only DPPH radical scavenging activity was showed in % whereas others were showed in “EC50”? Could the original materials exhibit the antioxidant? Did the authors check them?

- Table 5. Why is ACE inhibitory activity not expressed in IC50 like glucosidase inhibitory activity and amylase inhibitory activity? Also, could original materials exhibit the ACE inhibitory activity, glucosidase inhibitory activity, and amylase inhibitory activity? The authors were recommended to test the bioactivity of the original materials to confirm that those activities are from the products of hydrolysis.

- The conclusion should be written briefly.

- Line 413-416: there is two ref. 78. Please remove one.

Author Response

Dear Reviewer,

Thank you very much for your comments about our manuscript which were very useful and allowed to improve it. 

  • Line 41-42- Viscera were not used for the production of hydrolysates since the present study aims to formulate new seafood products that include hydrolysates with bioactive activities (SeaFood_Age project). On the other hand, our previous experience, and that of many other researchers, indicated that the organoleptic properties of FPH from viscera are not pleasant. Its taste is bitter, rancid and therefore not recommended to include in a food. Furthermore, the percentage of viscera in the case of fish discards used in this work ranged between 7-14%. Many fishes are discarded because they are below the legal size. For example, blue whiting and pouting were very small and viscera so small that it was not possible its physical separation from the fish body. 
  • Line 62-63 - The sentence was linked to the next paragraph
  • We have inserted the following paragraph to explain the reason for choosing Alcalase:

“The commerical endoprotease, alcalase, was applied for the production of fish hydrolysates due to its well-known high proteolytic activity and efficiency to digest a broad type of marine wastes.”

  • The subsections 2.6, 2.7, 2.8, etc were numbered.
  • Section 3.1

The comparison with the starting materials was not done because there is already a long experience, including the authors´ expertise on the preparation of fish protein hydrolysates, showing the occurrence of hydrolysis under the conditions followed in this work. The amount of naturally  present naturally peptides in fish is very limited and the peptide profile presented in this study  is due to the protein hydrolysis of raw material. The amino acid profile of each hydrolysate is a mixture of free amino acids of raw materials (a very small fraction) and those resultant from the hydrolysis of hydrolysate´s peptides. The composition of amino acids hydrolysates is important because it determines the physical and biological properties of hydrolysates.

Table 4- EC50/IC50 values are the hydrolysate concentration able to inhibit  50% of the radical activity or enzyme activity. A0.5 is the concentration of hydrolysate able to achieve an absorbance of 0.5 under the specific experimental conditions followed in evaluating the reducing power. AC0.5 was a typo error. The text was corrected.

In the case of DPPH, the percentage of inhibition increased with the increase of hydrolysate concentration, however, 50% of inhibition was not reached for any of the hydrolysates. So we chose the concentration of 3mg/ml to compare the DPPH radical scavenging activity of the different hydrolysates.

We did not check the DPPH radical scavenging activity of original materials. It is recognised that bioactive peptides usually consist of 2-30 amino acids and their activity is based on their composition and sequence. Few peptides exhibiting antioxidant activity are naturally present in fish and the overwhelming majority is incorporated in proteins but are inactive. Thus, they have to be released by the hydrolysis process to show their biological activities.

Table 5- We did not test different concentrations of hydrolysates in order to calculate the IC50 due to the high number of samples to be tested and the costs associated with this methodology. Once again, as the objective is to compare the biological activities of the different hydrolysates, we chose only 5mg/ml concentration to be tested. This concentration was selected based on the experience we have in evaluating this activity in other hydrolysates.

We did no test the ACE inhibitory activity of the original materials because it is well known that proteins do not exhibit this activity and the natural peptide concentration is very small as previously mentioned. This activity is exhibited by low molecular weight peptides (2-10 amino acids).

Conclusion- The conclusion was shortened as suggested.

Line 413-416 - The reference 78 was removed.

Kind regards,

Carla Pires

Round 2

Reviewer 1 Report

Please specify the parameters of table 1 are calculated on dry weight or wet weight.

Give the yields of hydrolysate fractions in Table 1. 

Author Response

Dear Reviewer,

Thank you very much for your comments.

The caption of table 1 was modified accordingly:

Table 1. Yield, chemical composition and molecular weight of dried FPH prepared with heads (He), skin+bones (Sb) and whole fish (Wh) of fish discards. Y: yield of FPH production, Mo: moisture, OM: organic matter, TP: total protein, TF: total fat, Dig: digestibility, Mn: number average molecular weight, Mw: average molecular weight.

and the hydrolysis yield was included.

Kind regards,

Carla Pires

We specify in table 1 that ated on dry weight or wet weight.

Give the yields of hydrolysate fractions in Table 1. 

Reviewer 2 Report

The authors have replied all the queries which I listed so I suggest this revised manuscript may be considered for publication.

Author Response

Dear Reviewer,

Thank your very much for your comments.

Kind regards,

Carla Pires

Round 3

Reviewer 1 Report

The authors have made the necessary corrections and I have no more questions.